# Contribution of Fertilizer, Density and Row Spacing Practices for Maize Yield and Efficiency Enhancement in Northeast China

**DOI:** 10.3390/plants11212985

**Published:** 2022-11-04

**Authors:** Lin Piao, Shiyu Zhang, Junyao Yan, Tianxu Xiang, Yang Chen, Ming Li, Wanrong Gu

**Affiliations:** 1College of Agriculture, Northeast Agricultural University, Harbin 150030, China; 2Maize Research Institute, Heilongjiang Academy of Agricultural Sciences, Harbin 150086, China; 3Heilongjiang Academy of land Reclamation Sciences, Harbin 150038, China; 4Heilongjiang Kenfeng Seed Industry Co., Ltd., Harbin 150090, China

**Keywords:** grain yield, fertilizer, planting density, row spacing, resources utilization efficiency

## Abstract

The research aimed to assess the contribution of fertilizer, density, and row spacing in integrated cultivation measures and identify their regulation mechanism on canopy architecture and factors in biomass accumulation in spring maize. Zhengdan 958 was used as the experimental material, and the optimum mode (OM) was identified based on a preliminary experiment, including the optimal fertilizer management, suitable plant density and wide-narrow row spacing, and dramatic yield performance (11,445.16 kg ha^−1^ in 2017). Then, the effects of these practices on maize canopy structure performance were analyzed using the omission factors design experiment in optimum mode (OM). Treatments were set as follows: without fertilization (OM-F), without density (OM-D), and without wide-narrow plant spacing (OM-S). The results showed that the contribution of fertilization was maximum (23.85%), the second was intensive planting (16.05%), which promoted nitrogen accumulation and transport in leaves and stems via increased leaf area index and dry matter accumulation around the anthesis simultaneously, elevating the radiation utilization efficiency of the canopy and allowing a higher grain weight to be obtained. Wide-narrow row spacing yield contribution is minimum among the measures (8.649%), which could regulate the leaf and radiation transmittance in the middle and bottom layer of the canopy, while increasing the nitrogen accumulation of leaves and stalks in the silking stage, then significantly enhance the nitrogen transport and the matter accumulation of maize after anthesis. Our results showed that fertilizer management and density were the essential practices for integrated cultivation mode for northeast China. Moreover, wide-narrow row planting was advocated if permitted, which could elevate the utilization efficiency of radiation to 1%, and the yield of more than 11,000 kg ha^−1^ was obtained in Northeast China.

## 1. Introduction

As a staple grain crop grown worldwide, maize (*Zea mays*, L.) has only reached 70% of the yield potential in major maize production regions [1,2,3]. In China, maize comprises 30.76% of the national grain sown area and 35.63% of the grain output [4]. However, the increase in grain yield of maize based on traditional agriculture has stagnated since 1995, and further yield increases must depend on optimizing the production system using recommended management practices with the full mechanization of modern agriculture strategies [5,6,7,8,9,10]. It is urgent to build a maize canopy with high yield and efficiency to reach the yield potential remanent [11,12].

Increasing plant density and adjusting rows spacing and fertilizer management are key measures for grain production operated by humans, as well as the necessary factors for constructing integrated cultivation mode when optimizing the canopy structure of spring maize. Previous studies have shown that the optimization mode possesses better synchrony between fertilizer supply and nutrition demand [7,8,9,10,11,12,13]. The more uniform the distribution of radiant in the canopy, the greater reduction in the competition of individual plants in the canopy for radiation; the canopy capacity will also be improved, elevating the underlying density and allowing the dramatic grain yield to be obtained [13,14,15]. The increasing grain yield is also mainly due to the transport of dry matter, including the production and distribution of photosynthate in spring maize after anthesis [16,17], influenced by planting density, fertilizer, rows spacing and other cultivation measures [18,19,20,21,22,23]. Generally, the appropriate fertilizer management and row spacing can effectively improve the canopy capacity and significantly increase the biological yield [24,25,26,27,28]. On the contrary, rows spacing sometimes does not match the density, or improper field management and fertilizer application can weaken the leaf photosynthetic capacity, which significantly affects grain yield formation [29,30]. 

There is a significant amount of evidence on the positive regulation mechanism of fertilization, plant density, rows spacing, and a combination of other measures on grain yield formation [8,9,20]. To some extent, we recommended an integrated mode based on that suggestion. China is currently the most significant consumer of N fertilizer, representing a 50% gap in utilization efficiency with other major maize-producing nations in the world. Heilongjiang Province is in the northern region of the Jilin maize belt, one of the three golden maize belts in the world, and is the most important spring maize production and planting base in China. The annual maize planting area and total output account for more than 30% of the whole country. Therefore, it is of great significance to continuously enhance the production level of maize in the region. At the same time, the experimental site is in the dominant maize producing areas of Heilongjiang Province, with precipitation of 484 mm, effective accumulated temperature of 1515 °C and radiation amount of 2381.6 MJ/m^2^ in the growing season. It has climatic conditions conducive to yield improvement in Heilongjiang Province, so it was selected as the experimental site. Farmers in this region have gradually accepted integrated cultivation mode, including some measures, but limited attention has been paid to the depletion of natural resources, resources utilization efficiency, and even the deterioration of the environment. Compared with the average climate conditions of the past 20 years, due to the high temperature and dry weather in the maize fulling stage, the precipitation in the experimental area in middle and late July, 2017 was 0, which had decreased by 77.79% compared with the average value. In 2018, accumulated temperature increased by 3.05%, precipitation increased by 21.88%, average daily high temperature and average daily low temperature increased by 9.74% and 1.23%, respectively, and sunshine duration decreased by 26.15%; on the contrary, there was a 0.58% and 58.38% increase in accumulated temperature and precipitation, while there was a 0.12% and 28.34% increase in the highest and lowest daily temperature, respectively. However, precipitation was more than 719.7 mm, and the sunshine duration was slightly less than 1019.3 h. Research on the yield contribution of each key measure in the integrated cultivation model and its regulation mechanism would be an important theoretical basis to further improve maize productivity and help formulate policies. Consequently, the first objective of this study was to identify the optimum integrated mode based on a gradually intensive planting density experiment, then to explore the regulation mechanism of these key cultivation techniques based on the deduct key factor (from optimum integrated mode) experiment designs. The effect of key cultivation measures on the maize canopy structure, biomass accumulation, grain yield contribution, and resources utilization efficiency will be investigated in this study.

## 2. Results

### 2.1. The Identified Optimum Mode

Grain yield showed a parabolic tendency with the increase in plant density and tended to be higher between 70,000 plants ha^−1^ and 80,000 plants ha^−1^. The wide and narrow spacing planting model could increase the yield compared with dense planting (Figure 1a,b). In addition, dense planting and narrow-width planting could significantly change canopy biomass accumulation (Figure 1c,d) light interception (Figure 1e,f), which could promote maize growth and grain yield accumulation. Future research results suggest that suitable cultivation technique matching was beneficial to achieve high yield and high efficiency of spring maize in the cold region of China. The plant density could at least increase to 80,000 plants ha^−1^.

### 2.2. The Vertical Distribution of LAI

The vertical distribution of leaf area index (LAI) in the canopy was increased rapidly, and then decreased with the increase in plant height, including the maximum point located at 180–240 cm. Compared with the OM treatment, the upper LAI percentage of other treatments was reduced remarkably. Fertilization could increase the LAI (especially > 180 cm: significantly) in the canopy of maize. Increases in the density could increase the leaf area index (LAI) in the middle and lower layer of maize (at 180 cm and 120 cm), and row spacing treatments had a significant effect on the leaf area index (LAI) of 180 cm layer in 2018. On the other hand, in 2019, row spacing had a significant effect on the leaf area index of the middle. The leaf area index at 120–180 cm under uniform row spacing treatment was significantly higher than that in narrow-width row treatment, which showed that the wide and narrow row planting, density plant, and fertilizer cultivation measures have a significant effect on the structure of the maize canopy leaves (Figure 2).

### 2.3. The Vertical Distribution of Leaves Weight and Transmittance

There were significant differences in leaf biomass among treatments. Compared with the treatments without fertilization, the leaves biomass of each plant in the fertilization (OM, OM-D, OM-S) canopy were relatively large, and the optimal model (OM) had a larger leaf dry matter weight. In the low-density population, the leaves of each plant were relatively luxuriant; however, the dry matter weight of leaves was significantly lower than that of OM treatment due to the limitation of population plants number. Comparing traditional planting with uniform row spacing, narrow-narrow-row planting (OM, OM-F, OM-D) significantly reduced the dry matter distribution of leaves at or below ear level in the canopy. The increase in planting density and fertilizer mainly affected the distribution of leaf and light in the maize canopy by increasing leaf dry weight and light transmittance at and above ear position. However, the change in row spacing affected the distribution of leaf and light at ear position and below (Figure 3).

### 2.4. The Vertical Distribution of Nitrogen Content and Translocation

The maize canopy can be divided into three sections around the ear and leaf nitrogen content increases gradually with the increase in canopy height. Nitrogen content in the stem increased with the increase in canopy height also under no fertilizer treatment in 2017 (increasing first and then decreasing in 2018), while nitrogen content in the stem showed upper layer > lower layer > middle layer under fertilization treatment. The results showed that nitrogen fertilizer had a significant effect on the nitrogen content of maize stalk (Figure 4).

The analysis of nitrogen translocation amount, translocation rate, and its contribution to grain in 2019 showed the increase in fertilizer significantly promoted nitrogen translocation amounts of maize leaves and stem translocation amount by 64.7% and 73.1%, respectively, decreased leaf translocation rate of maize, and increased stem translocation rate. The contribution rate of nitrogen translocation to grain of maize under no fertilizer treatment was slightly higher than that under fertilizer treatment. Increasing planting density decreased the stalk N translocation rate by 0.27% and increased leaf translocation by 3.58%. The contribution rate of stem nitrogen translocation to grain under low density was higher than that of under high density. Row spacing had significant effects on N translocation, translocation rate and contribution rate of maize stem, and wide and narrow row spacing could effectively improve N translocation of maize stem (Table 1).

### 2.5. The Accumulation of Dry Matter and Translocation

Cultivation measures had significant effects on dry matter accumulation. Increasing fertilizer and planting density could significantly increase dry matter accumulation before and after flowering by 29% and 30%, 20%, and 24%, respectively. Wide and narrow row treatment could promote dry matter accumulation before flowering, and uniform row spacing treatment significantly increased dry matter accumulation after flowering. There were no significant differences between fertilization and planting density on dry matter translocation, translocation rate and contribution rate of translocation to yield, but the grain yield formation under these treatments much depends on the dry matter translocation of vegetation organ. Density had the greatest effect on dry matter accumulation of maize. Therefore, increasing planting density was an important measure to improve the dry matter accumulation of maize (Table 2).

### 2.6. The Grain Yield and Resource Utilization of Maize

Different cultivation measures had significant effects on maize yield, and the yield was OM > OM-S > OM-D > OM-F. Fertilizer and density measures significantly affected maize yield, and the technical contribution rate was 23.85% and 16.05%, respectively. The yield of fertilization was significantly higher than that of no fertilizer, and the yield of high density was significantly higher than that of low-density treatment. Fertilization significantly increased the 100-kernel weight and grain yield of maize. When the density increased, the number of rows and the number of grains per row decreased, while the 100-grain weight and yield increased. The yield of high-density treatment was significantly higher than that of low-density treatment. Canopy resources use efficiency was consistent with the trend of yield among treatments. Cultivation practices improve its canopy leaf distribution, nitrogen content and light interception, the accumulation of biomass and the growth, resulting in the high efficiency of nitrogen and heat resource utilization, kernel number and grain weight (Table 3).

## 3. Discussion

To meet the rigid growing global demand, the production of cereal crops should at least increase by 70% by 2050 [31,32,33,34]. Considering the current maize yield per unit area, it will be challenging to meet such yield demand, and continuous yield improvement is still the key issue of current and future maize cultivation management research [35,36]. Studies have shown that at least 50% of the increase in maize yield over the last century can be attributed to the improvements in cultivation measures [37,38,39]. Among them, the density increase in the maize canopy is the most critical measure for high yield in a large area [40]. In order to achieve intensive maize canopy and in future to construct the integrated measures model with optimized yield and efficiency, four density gradients of 60,000–100,000 plants ha^−1^ were set in this study. Similar to results from previous studies compared with uniform spacing, we obtained the highest yield at 80,000 plants ha^−1^ under wide-narrow plant spacing in Heilongjiang Province of China. Therefore, we set plant density at 80,000 plants ha^−1^ to demonstrate the regulation mechanism of key measures for canopy structure optimization with high efficiency and yield, and then quantified the yield contribution of cultivation measures. Moreover, the optimization canopy structure model was constructed to reduce the LAI proportion of the lower layer. Meanwhile, the proportion of the middle and upper layer should be more than 84%, including the accounts of the upper layer, which must be about 26–30%. This model is beneficial to maize intensive planting and high yield, improves grain benefits, has greater production potential, and has research value and guiding significance for realizing high maize yield in cold agroecological regions such as Heilongjiang province.

The fertilization strategy includes the timing, method, and amount of fertilizer that can influence the final grain yield and in turn affect the plants growth, leaf color, leaf matter production and biomass accumulation and transport [41]. Nitrogen is a movable nutrient in the plant, and nitrogen deficiency was first expressed in the mature tissues and then accelerated the senescence of leaves. The nitrogen supply amount should be different regarding varieties, which can be up to 30%. Therefore, the application amount of nitrogen should be considered for different varieties. This study shows that suitable fertilization can significantly increase the LAI of the upper layer of the ear in the silking stage, improve the canopy structure in the canopy, maintain the higher functional period of leaves, and reduce the radiation loss in the canopy. Three measures improved the utilization rate of radiation and accumulated temperature resources effectively, increased dry matter accumulation and nitrogen accumulation around the silking stage and promoted dry matter transport. In this experiment, fertilizer had little effect on the number of grains per ear, which was slightly different from previous results that suggest the same relationship with the high nitrogen absorption ability of Zhengdan 958, heavy precipitation, little radiation during the panicle differentiation stage, and the nutrient of field soil [42]. Planting density per hectare increased from 60,000 to 80,000 in Northeast China and could significantly increase the leaf area index in maize growth period, extend the leaves photosynthetic time, and enhance the difference in leaf area between the upper layers and decrease canopy lower (< 120 cm) layer light transmittance, which significantly improves the efficiency of maize nitrogen physiological utilization and nitrogen agronomy. Dry matter accumulation and transport in the stem were increased after silking. By increasing the utilization rate of radiation and accumulated temperature resources, the yield contribution rate was 16.05%. Additionally, previous research has successfully considered the interaction between fertilization and plant density, which could enhance the dry matter accumulation of the canopy then increase the grain yield and the efficiency of resource utilization simultaneously [32,43,44]. By adjusting row spacing, the ventilation and light transmittance environment in the maize canopy can be improved [13,17,45,46]. 

Wide-narrow spacing treatment can improve light interception in the middle and lower layer of the canopy, increase leaf area and radiation utilization rate, postpone the leaf senescence in late growth period, promote dry matter accumulation, and improve 100-grain weight and row number of ears, which is conducive to the increase in yield [13,14,47,48]. However, studies have shown that uniform row spacing treatment has relatively high nitrogen accumulation [32], and wide-narrow spacing and uniform spacing cultivation modes have no significant difference in yield [49], even leading to lower yield if the wide row is larger [50]. This study shows that leaf area index in the wide-narrow spacing (40 cm + 90 cm) middle layer (120–180 cm) was significantly smaller than uniform spacing, improving canopy light distribution and increasing the light transmittance of the ear layer (Figure 5) and the photosynthetic product transport after flowering; the yield contribution rate was 8.649%. The increase in grain yield under the integrated cultivation measures mainly depended on the regulation of the key cultivation and interaction measures [10,51]. In the cold agroecological region of China, we may set a fertilizer management strategy based on the characteristics of soil and climate and variety first, then determine the plant density and consider the wide-narrow spacing cultivation in the allow conditions in order to maximize the efficiency of resources and population grain yield (Figure 6).

## 4. Materials and Methods

### 4.1. Experimental Design and Field Management

The research was conducted at Xiangfang Farm Agricultural Experimental Station in 2017, and Agricultural Experimental Station of Northeast Agricultural University at Xiangfang District Chenggaozi town, Minsheng village in 2018 to 2019. The soil texture is black loam (source of soil classification: ISS-CAS 2003) and the previous crop is maize. The basic characteristics of the soil under topsoil 20 cm are listed in Table 4. Maize hybrid “Zhengdan 958” (Henan Academy of Agricultural Sciences) was chosen as the experimental material and sown in two factors (the wide-narrow plant spacing and density 60,000–100,000 plants ha^−1^) split zone designs, with three replicates and omission experimental design. Wide-narrow rows space (40 + 90 cm) 80,000 plants ha^−1^ density under fertilization condition was sat as the control treatment (OM), three omission experimental treatments were set as without fertilization (OM-F), without density (OM-D) and without wide-narrow plant spacing (OM-S), respectively (Table 5 and Figure 7). In late April, the seeds were planted with handle and agricultural fertilizer (N:P:K = 15:15:15; 375 kg ha^−1^) containing 56.25 kg ha^−1^ of N, P_2_O_5_, and K_2_O. An additional side-dressing of 138 kg ha^−1^ N was applied at the elongation stage in 2017, and 140 N kg ha^−1^, 55 P_2_O_5_ kg ha^−1^, 70 K_2_O kg ha^−1^ (N:P:K = 28:11:14; 500 kg ha^−1^) in late April in 2018 and 2019. Then, side-dressing was applied 69 N kg ha^−1^ and 35 K_2_O kg ha^−1^ at the elongation stage. The row spacing was 0.65 m for uniform spacing and 0.9 + 0.4 m for wide-narrow rows spacing. Plant density 80,000 plants ha^−1^ for OM, OM-F, OM-S, and 60,000 plant ha^−1^ were used for OM-D treatment. All treatments had three replicates of 5 m long by 3.9 m wide plots in 2017–2018 and three replicates of 5 m long by 5.2 m wide plots in 2019. Diseases and pests were well controlled throughout the experiment (Figure 8).

### 4.2. Determination of Biomass Accumulation

Three plant samples with the same growth were selected in the plot for the dry matter accumulation measurement. In maize V8 (8 leaves with visible leaf collars), V12 (12 leaves with visible leaf collars), R1 (flowering), R3 (milking), and R6 (physiological maturity) stages, respectively, in 2017–2019, the samples were fixated at 105 °C for 30 min, and then dried at 75 °C to a constant weight [20,48]. Moreover, 2 m^2^ area plants in each treatment plot were selected as samples at the filling stage to survey the vertical canopy structure. Looping pliers were used to cut and sample in layers from the ground to plant top in a 30 cm layer (Figure 9). Furthermore, leaves and stems were stored separately and fixated at 105 °C for 30 min, and then dried at 75 °C to constant weight.

### 4.3. Determination of Leaf Area Index

Five representative maize plants were selected in each treatment plot in V5 stage, and the length and width of leaves were determined to calculate the leaf area index using the following formula:(1)LA=L×W×a
(2) LAI=LA/PA 
where *LAI* is the leaves area index, *L* is the length of the green part and *W* is the maximum width of each individual leaf belonging to one plant, and *LA* and *PA* are the leaf area and plant area of one plant repeated 3 times in each treatment plot. Letter “*a*” is the leaf area coefficient (0.5 for non-expanded but visible leaves or leaves’ tip part and 0.75 for expanded leaves).

### 4.4. Determination of Population Transmittance

On a cloudy day during the silking period, measurements were made in layers with a stick illumination meter (GZ-1 type) of 30 cm, and the directions along the ridges and across ridges were determined, respectively. The directions along the ridges of wide and narrow rows were divided into wide rows and narrow rows. Moreover, a CI-110 canopy digital image analyzer was used to capture the hemisphere images of the ear and bottom layers of the maize population [20,48].

### 4.5. Determination of Nitrogen Content and Nitrogen Efficiency

The samples of biomass accumulation into power for nitrogen content were determined by the semi-micro Kjeldahl method [22] and calculated using the following formula: (3)NAA=NC×W
(4)$$NTA=NAA_{\left(silking\right)}-NAA_{\left(maturity\right)}$$
(5)$$NTE=\frac{NTA}{NAA}\times 100\%$$
(6)$$NTCR=\frac{NTA}{NAA_{\left(grain\right)}}\times 100\%$$
(7)$$NPE=\frac{GY_{\left(N\;application\right)}-GY_{\left(without\;N\right)}}{NAA_{\left(N\;application\right)}-NAA_{\left(without\;N\right)}}\times 100\%$$
(8)$$NAE=\frac{GY_{\left(N\;application\right)}-GY_{\left(without\;N\right)}}{NAA_{fertilizer\;application\;amount}}$$
(9)$$NUE=\frac{GY}{NAA_{\left(total\right)}}$$
where *NAA*, *NTA*, *NTE*, *NCRT*, *NPE*, *NAE*, *NUE* are nitrogen accumulation amount (g), nitrogen translocation amount (g), nitrogen translocation efficiency (%), nitrogen translocation contribution to grain nitrogen accumulation amount (%), nitrogen physiological efficiency (kg kg^−1^), nitrogen agriculture efficiency (kg kg^−1^), and nitrogen using efficiency (kg kg^−1^), respectively, and *NC* (%), *W* (g), and *GY* (kg ha^−1^) were nitrogen content, dry matter weight of each organ (for example leaf, stem, and grain in different growth stage such as silking and maturity), and grain yield, respectively [23,31,32,52,53]. The *NAA* in Formulas (7) and (9) were the total (including leaf, stem, and grain) nitrogen accumulation amount in maturity stage.

### 4.6. Determination of Population Grain Yield of Maize

Maize was hand-harvested after physiological maturity. At harvest, maize was collected from each plot separately, and then 30 ears were randomly chosen to record ear rows and grain number of two opposite rows of each ear. The final yield was standardized at 15.5% moisture. Three replicates of 100 kernels were randomly sampled from the dried grain, weighed, and averaged to determine the 100-kernel weight.

### 4.7. Determination of Radiation and Accumulated Temperature Utilization

(10)$$RUE=H\times \frac{GY}{SRs}$$(11)$$GDD_{S}=\sum_{0}^{n}\frac{T_{min}+T_{max}}{2}-T_{base}$$(12)$$GUE=\frac{GY}{GDDs}$$
where *H* is the combustion heat of dry matter per unit area (J R^−1^), and the value is 1779 × 10^4^ J kg^−1^; *SRs* is the total solar radiation during the whole growth period (*SRs*: MJ m^−2^). *GDDs* is the total accumulated temperature of the whole maize growth stage (*GDDs*: °Cd; *T_min_* and *T_max_* are the min temperature and max temperature of a day; the value of *T_base_* is 10 °C) during the whole growth period, and *GY* is the crop yield (kg ha^−1^) [33].

### 4.8. Data Analysis

SPSS 20.0 (SPSS, Statistics, Chicago, IL, USA) was used for the analysis of variance and differences were compared at the 0.05 probability level between the means; LAI, dry matter weight, nitrogen content and transmittance at different layers and dry matter accumulation dynamics during different periods were conducted in SigmaPlot 12.5 (Systat Software Inc., San Jose, CA, USA).

## 5. Conclusions

In cold agroecological regions of China, the optimization integrated cultivation measures should include density planting, applying fertilizer based on soil testing to meet the growth requirements of maize plants, and wide-narrow rows spacing for obtaining high yield and efficiency. Canopy structure and radiation distribution were the primary factors of regulating the mechanisms of the measures to ensure LAI maintenance and the photosynthetic capacity of the maize canopy after silking, and improve the light interception of the canopy middle and bottom layer, thus improving the utilization rate of light energy to 1%, so that a grain yield above 11,000 kg ha^−1^ can be obtained.

## Figures and Tables

**Figure 1 plants-11-02985-f001:**
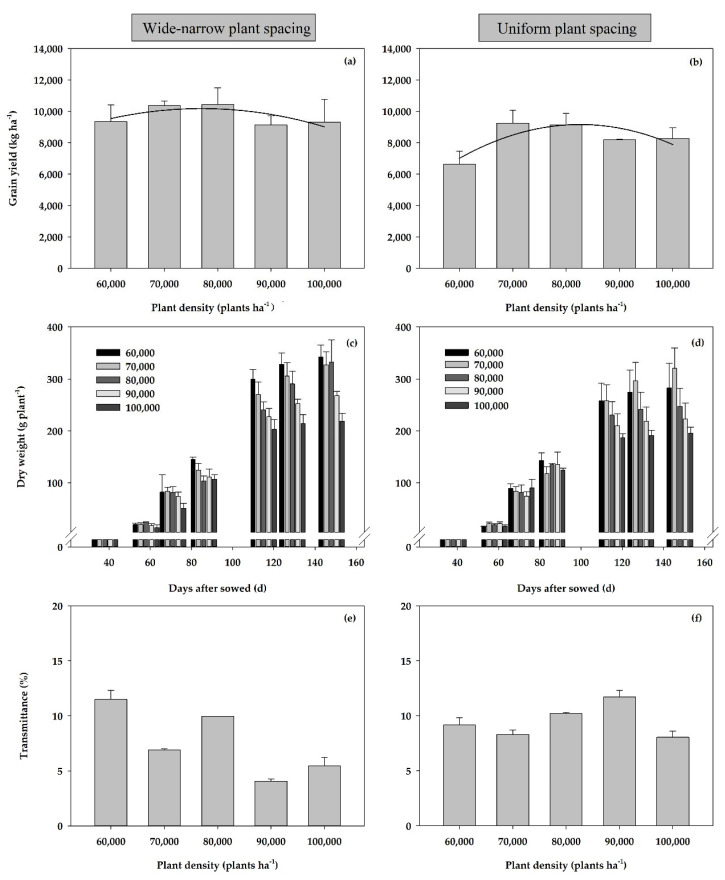
Dynamics of dry matter weight, transmittance of canopy and variation of yield with plant density under wide-narrow (**a**,**c**,**e**) and uniform plant spacing (**b**,**d**,**f**).

**Figure 2 plants-11-02985-f002:**
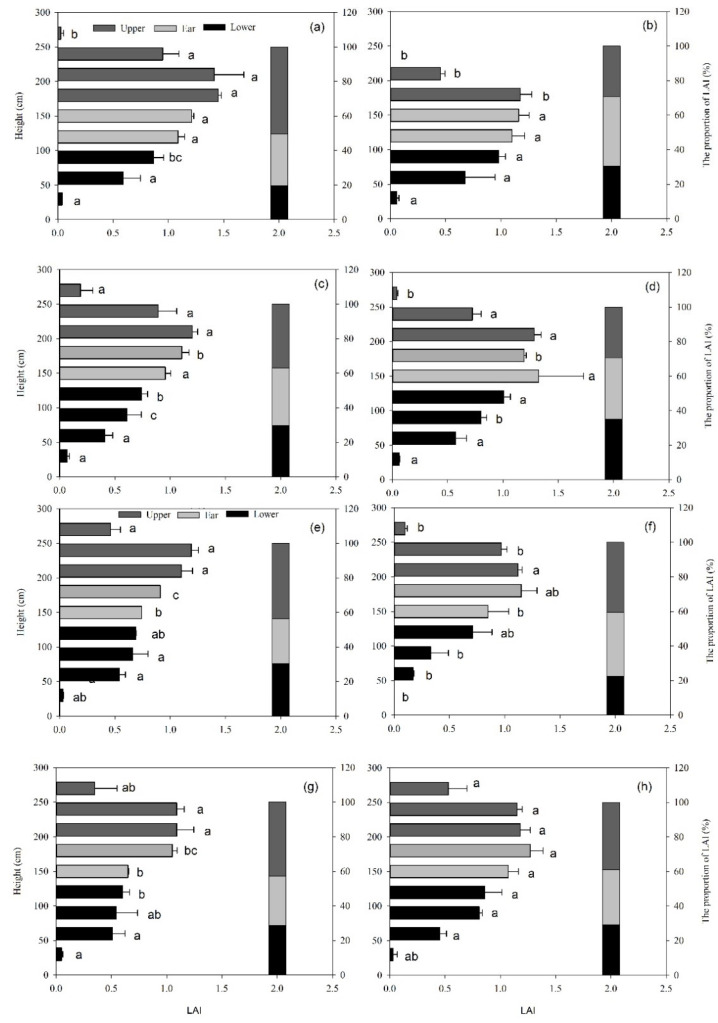
The vertical distribution of maize leaf area index (LAI) under difference treatments, respectively, and the data was all collected in 2018 (**a**–**d**) and 2019 (**e**–**h**). Sub-graphs (**a**) and (**e**) indicate treatment OM; sub-graphs (**b**) and (**f**) indicate treatment OM-F; sub-graphs (**c**) and (**g**) indicate treatment OM-D; sub-graphs (**d**) and (**h**) indicate treatment OM-S. The gray bar represents the LAI of ear layers, dark gray and break bars represent the LAI of upper and lower ear layers respectively. Different small litters indicate significantly different at 0.05 probability level among the treatments.

**Figure 3 plants-11-02985-f003:**
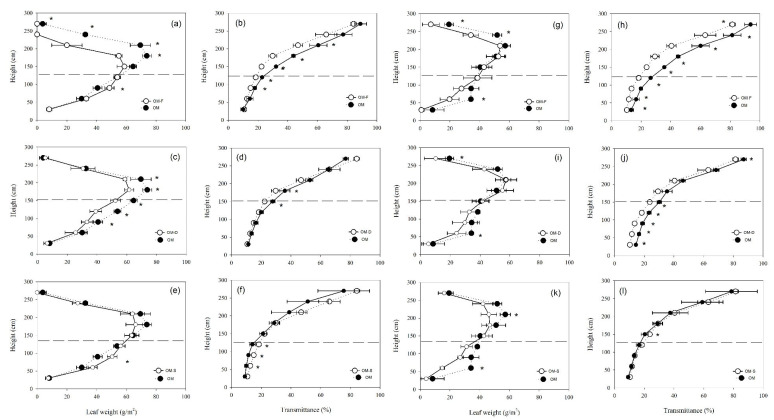
The vertical distribution of maize leaf weight and transmittance under different treatments, and the data was all collected in 2018 (sub-graphs (**a**–**f**)) and 2019 (sub-graphs (**g**–**l**)). Dotted line indicates the height of the ear; “*” represent significance at the 0.05 probability.

**Figure 4 plants-11-02985-f004:**
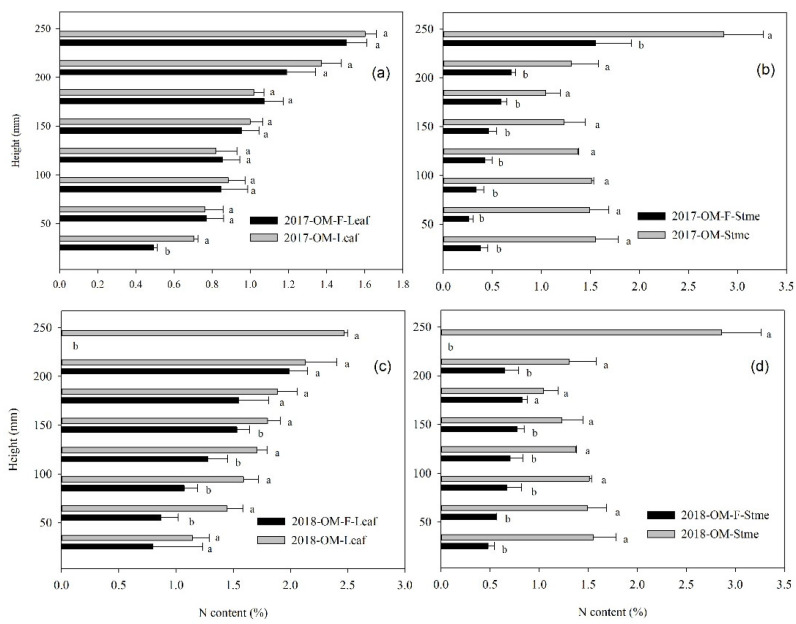
The vertical distribution of N content of maize leaf (**a**,**c**) and stem (**b**,**d**) at silking stage in 2017 (**a**,**b**) and 2018 (**c**,**d**), respectively. “a, b” indicate significantly different at the 5% probability level.

**Figure 5 plants-11-02985-f005:**
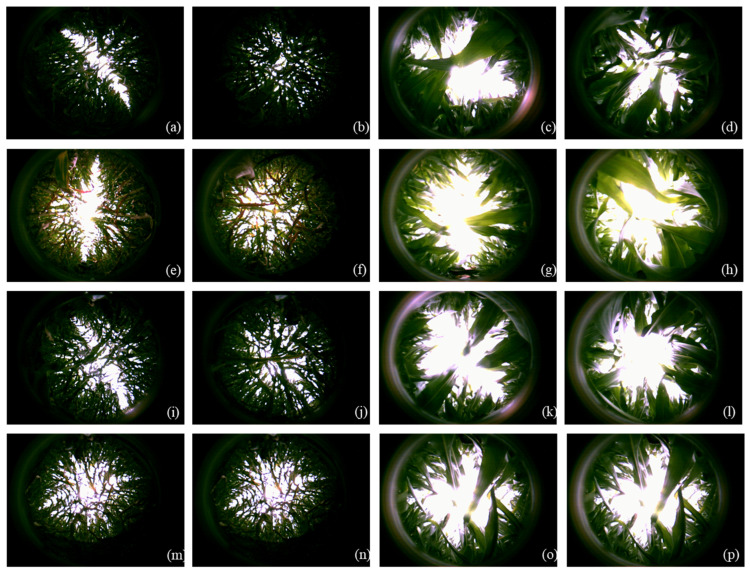
The hemisphere images of ear layer and bottom layer of maize population. (**a**–**d**) is the optimum mode (OM; CK); (**e**–**h**) is OM treatment without fertilization (OM-F); (**i**–**l**) is OM treatment without density (OM-D); (**m**–**p**) is OM treatment without wide-narrow plant spacing (OM-S; because the uniform row spacing plant, the hemisphere images of wide and narrow row were the same, images (**m**,**n**); (**o**,**p**)). And (**a**,**e**,**i**,**m**) is wide row bottom layer; (**b**,**f**,**j**,**n**) is narrow row bottom layer; (**c**,**g**,**k**,**o**) is wide row ear layer; (**d**,**h**,**l**,**p**) is narrow row ear layer.

**Figure 6 plants-11-02985-f006:**
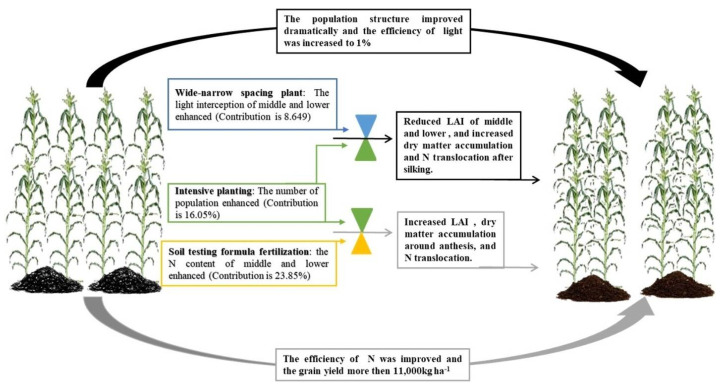
The regulatory mechanisms of fertilization, density, and wide-narrow spacing plant on maize population in Northeast China.

**Figure 7 plants-11-02985-f007:**
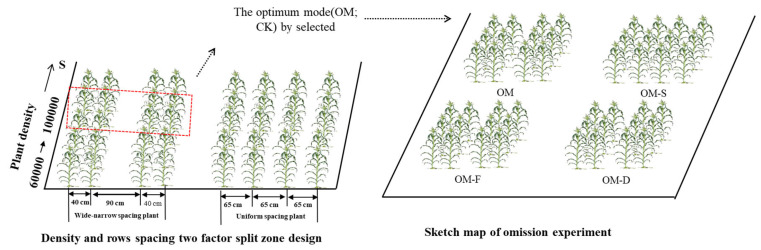
The sketch map of field experiment in 2017 to 2019, Based on the two factor split zone experiments, we identified the optimum mode (OM) as control treatment for omission experiment. The red box was used to show that the plant density (80,000 plants ha^−1^) and plant spacing (wide-narrow plant spacing) of optimum mode (OM treatment) was selected based on the different plant density and plant spacing tests in the left sub-graph.

**Figure 8 plants-11-02985-f008:**
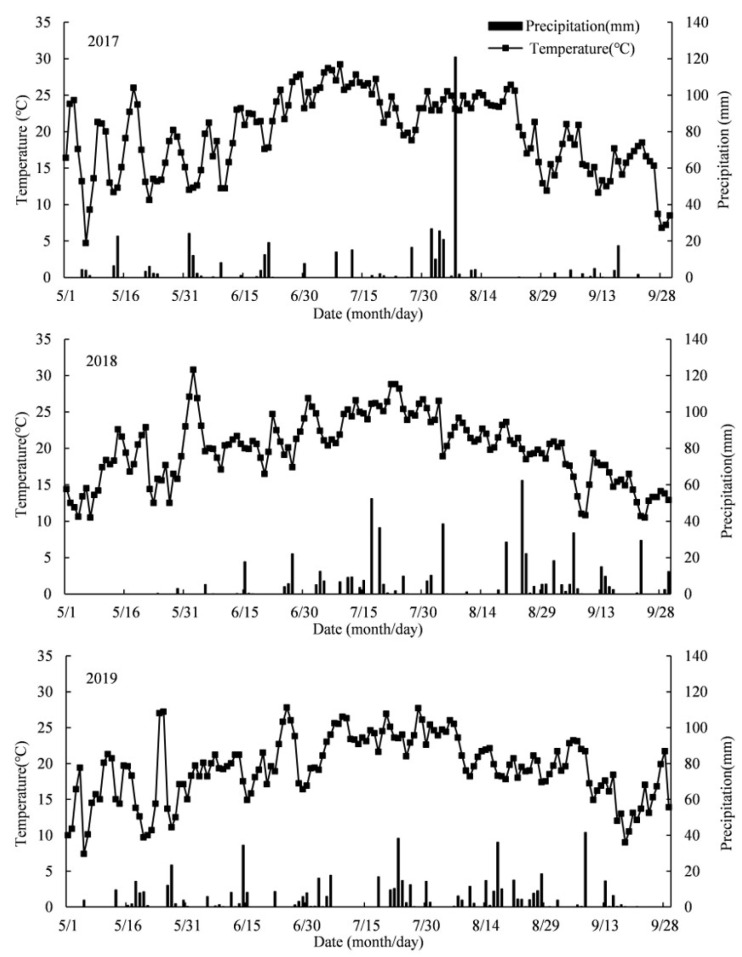
Monthly rainfall distribution and mean temperature during spring maize growing stage in 2017, 2018 and 2019.

**Figure 9 plants-11-02985-f009:**
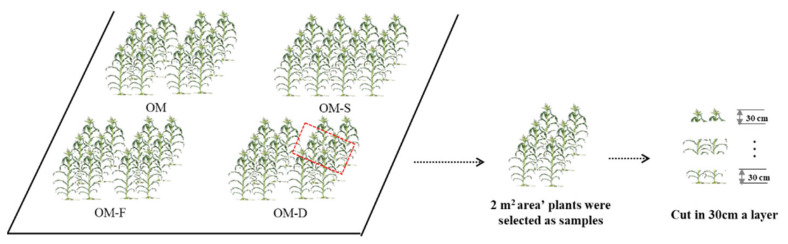
The sketch map of the samples collection method of canopy vertical structure. 2 m^2^ area’ plants were selected as samples and cut in layers from the ground to plant top in 30 cm a layer. The red box was used to show that the 2 m^2^ area’ plants was selected as samples from different treatments in the left sub-graph.

**Table 1 plants-11-02985-t001:** Effects of different cultivation modes on nitrogen transport in maize in 2019.

Treatments	Nitrogen Content (Maturity; %)		Translocation Amount (kg ha^−1^)		Translocation Rate (%)		Contribution Ratio (%)	
	Leaf	Stem	Leaf	Stem	Leaf	Stem	Leaf	Stem
OM-F	0.442 ^b^	0.15 ^b^	16.84 ^b^	15.11 ^a^	53.30 ^a^	60.07 ^a^	29.33 ^a^	26.90 ^a^
OM-D	0.72 ^a^	0.261 ^a^	27.14 ^a^	28.39 ^a^	46.30 ^a^	61.72 ^a^	28.08 ^a^	28.44 ^a^
OM	0.667 ^a^	0.279 ^a^	27.74 ^a^	26.16 ^a^	50.88 ^a^	61.45 ^a^	29.17 ^a^	25.96 ^a^
OM-S	0.662 ^a^	0.335 ^a^	21.49 ^ab^	18.66 ^b^	44.91 ^a^	49.80 ^a^	24.05 ^a^	21.14 ^a^

Note: Values were the mean of the data collected in 2019, and followed by different small letters within a column indicate significantly different at the 5% probability level (Duncan; *n* = 3).

**Table 2 plants-11-02985-t002:** Effect of different cultivation measures on dry matter accumulation and transport of maize.

Treatments	DMA (kg ha^−1^)		AR (%)		TA (kg ha^−1^)	TR (%)	CR (%)
	Before Silking	After Silking	Before Silking	After Silking			
OM-F	7789.2 ^b^	6057.73 ^c^	48.61 ^a^	37.64 ^b^	1731 ^a^	22.1 ^a^	20.17 ^a^
OM-D	8362.3 ^b^	6366 ^c^	48.33 ^a^	36.78 ^b^	1996 ^a^	23.84 ^a^	20.05 ^a^
OM	10,082 ^a^	7900.67 ^b^	50.1 ^a^	39.29 ^ab^	2181 ^a^	21.37 ^a^	18.5 ^a^
OM-S	9969.53 ^a^	9222.13 ^a^	45.6 ^a^	42.1 ^a^	747.4 ^b^	7.72 ^b^	6.57 ^b^

Note: DMA, dry matter accumulation of plant (not include grain weight); AR, accumulation rate, which was computed with dry matter accumulation (before or after silking) to the dry matter accumulation of plant (include grain weight) at maturity; TA, translocation amount, the different values of DMA between before and after silking; TR, translocation rate, which was computed with TA to DMA before silking; CR, contribution ratio, which was computed with TA to the grain yield of the treatment. Values were the mean of 2018 and 2019 and followed by different small letters within a column indicate significantly different at the 5% probability level (Duncan; *n* = 6).

**Table 3 plants-11-02985-t003:** Effect of different cultivation measures on grain yield and resource utilization of maize.

Treatments	Grain Yield (kg ha^−1^)	Rows (No ear^−1^)	Kernels (No row^−1^)	Kernels (No ear^−1^)	100-Kernel Weight (g)	RUE (%)	GUE (kg °C^−1^ d^−1^)	NUE (kg kg^−1^)	NAE (kg kg^−1^)	PFPN (kg kg^−1^)	NPE (kg kg^−1^)	MCR (%)
OM-F	9365 ^b^	14.80 ^b^	34.56 ^a^	511.9 ^a^	22.43 ^a^	0.69 ^b^	3.17 ^b^	-	-	41.20 ^c^	-	23.85
OM-D	9994 ^b^	15.47 ^ab^	36.65 ^a^	546.5 ^a^	24.96 ^a^	0.75 ^ab^	3.40 ^b^	17.58 ^a^	7.30 ^b^	48.12 ^b^	21.40 ^b^	16.05
OM	11,598 ^a^	14.82 ^b^	35.29 ^a^	543.6 ^a^	25.80 ^a^	0.87 ^a^	3.94 ^a^	17.16 ^a^	16.3 ^a^	56.83 ^a^	41.66 ^a^	-
OM-S	10,675 ^ab^	15.00 ^a^	33.60 ^a^	504.2 ^a^	23.79 ^a^	0.8 ^ab^	3.62 ^ab^	17.32 ^a^	12.75 ^ab^	53.44 ^a^	36.53 ^ab^	8.649

Abbreviation: RUE, radiation use efficiency; GUE, growth degree days use efficiency; NUE, nitrogen use efficiency; NPE, nitrogen physiological utilization efficiency; NAE, nitrogen agriculture efficiency; PFPN, nitrogen partial factor productivity; MCR, measures contribution ratio. Note: Values were the mean of 2017 to 2019 (the values of NUE and NPE were calculated based the data of 2019), and followed by different small letters within a column indicate significantly different at the 5% probability level (Duncan; *n* = 9 or *n* = 3 for NUE and NPE).

**Table 4 plants-11-02985-t004:** Soil basic fertility characteristics.

Year	pH	Organic Matter (g kg^−1^)	Total Nitrogen (mg kg^−1^)	Available Nitrogen (mg kg^−1^)	Available Phosphorus (mg kg^−1^)	Available Potassium (mg kg^−1^)
2017	6.51	29.24	104.74	103.52	73.43	210.75
2018–2019	6.80	26.30	98.00	190.00	82.22	123.10

**Table 5 plants-11-02985-t005:** Treatment design for fertilization, planting density and row spacing.

Treatment	N Fertilizer kg ha^−1^	P Fertilizer kg ha^−1^	K Fertilizer kg ha^−1^	Planting Density (Plants ha^−1^)	Row Spacing (mm)
Wide-narrow spacing	194.25	56.25	56.25	60,000–100,000	90 + 40
Uniform spacing	194.25	56.25	56.25	60,000–100,000	65
OM(CK)	209	55	105	80,000	90 + 40
OM-F	0	0	0	80,000	90 + 40
OM-D	209	55	105	60,000	90 + 40
OM-S	209	55	105	80,000	65

## Data Availability

The data presented in this study are available within the article.

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
