# Peer review of "Contribution of Fertilizer, Density and Row Spacing Practices for Maize Yield and Efficiency Enhancement in Northeast China"

_plants, 2022, doi:10.3390/plants11212985_

Round 1

Reviewer 1 Report (New Reviewer)

Fertilizer, density, and row spacing management are the main agronomic methods to regulate maize growth and yield. Previous reports only emphasized the single measure or measure optimization, without considering the priority sequence among these managements. How to choose to reduce the loss of yield, there is still a lack of relevant research reports. The research aimed to assess the contribution of fertilizer, density, and row spacing in the integrated cultivation measures and identify their regulation mechanism on canopy architecture and factors in biomass accumulation in spring maize. Because of this, the current interesting study is on a topic of relevance and general interest to the readers of the journal. The study was conducted in a field that was fertilizer, density, and row spacing management. The design of the field with three-year period of the experimental treatment makes the dataset seem quite useful for the purpose. I think the manuscript to be overall well written and much of it to be well described. I felt confident that the authors performed careful and thorough field and data processing. Therefore, I recommend that a minor revision is warranted.

Minor comments:

1. It is suggested to modify the color marks in the manuscript for clearer expression at first time.

2. Line 47-49: “Previous studies have shown that the optimization mode possessed the better synchrony between fertilizer supply and nutrition demand.” Please add some references for better description.

3. Page 16:The sentence: “Figure 7. The sketch map of the samples collection method of canopy vertical structure.” Figure 7 should be changed to Figure 9.

4. The conclusion should focus on the main points without explanation or description.

5. Reference 15 and 17:Of should be changed to “of”. The author is requested to carefully revise the format of the references.

Author Response

Minor comments:

  1. It is suggested to modify the color marks in the manuscript for clearer expression at first time.

Reply:Thank you for your suggestion and we have revised the MS according the journal requirements.

  1. Line 47-49: “Previous studies have shown that the optimization mode possessed the better synchrony between fertilizer supply and nutrition demand.” Please add some references for better description.

Reply:According to your suggestion, we have added some references to better support our statement.

  1. Page 16:The sentence: “Figure 7. The sketch map of the samples collection method of canopy vertical structure.” Figure 7 should be changed to Figure 9.

Reply:Very sorry for mistake and we have changed all Figures and Tables. Thanks a lot.

  1. The conclusion should focus on the main points without explanation or description.

Reply:Yes, you are right and we have refined the conclusion.

  1. Reference 15 and 17:Of should be changed to “of”. The author is requested to carefully revise the format of the references.

Reply:We have changed this mistake and thanks a lot.

Reviewer 2 Report (New Reviewer)

Further exploration of the high yield potential of maize must depend on a reasonable comprehensive cultivation model. Through three years of field experiments, this study had obtained convincing and consistent data. This experiment aimed to deeply analyze the contribution of population function characteristics to yield formation under the multiple measures model (Fertilizer, density, and row spacing), which was of great significance for improving the field canopy phenotype theory of spring maize and constructing the high yield and high efficiency optimal cultivation model in Northeast China. The revised comments were followed.

(1)Abstract: “Hence, we suggest…” may be changed to “Our results showed that……”

(2)Keywords: Delete “cultivation measures” for better clarity.

(3)Introduction:

Line 65 “in the world,” should changed to “in the world.” The author is requested to revise the language details of punctuation marks and similar errors throughout the article.

Line 61-62, Please quote two references in this sentence “There was lots of evidence of the positive regulation……..”

(4) Results:

The lower case letters in the table 1-3 should be superscript.

(5) Discussion:

Line 224-225: Delete the sentence “maximum yield at 70,000 plants ha-1 “

(6) Materials and methods

Please cite two references in 4.2 and 4.4.

Please pay attention to the case of the first letter of the English word in the sentence. Such as ”4.7. Determination of radiation and accumulated temperature Utilization”.

Line 344-345: Figure 7 should be changed to Figure 9.

(7)References need to be uniformly revised and improved according to the “Plants”.

Author Response

(1)Abstract: “Hence, we suggest…” may be changed to “Our results showed that……”

Reply:Thank you for your suggestion and we have revised the abstract.

(2)Keywords: Delete “cultivation measures” for better clarity.

Reply:According to your suggestion, we have deleted " cultivation measures " for clearer expression.

(3)Introduction:

Line 65 “in the world,” should changed to “in the world.” The author is requested to revise the language details of punctuation marks and similar errors throughout the article.

Reply:We are very sorry that our mistakes have caused obstacles when reading MS. We have revised and improved the language and grammar of the MS. We look forward to your further guidance.

Line 61-62, Please quote two references in this sentence “There was lots of evidence of the positive regulation……..”

Reply:According to your suggestion, we have added two references to better support our statement.

(4) Results:

The lower case letters in the table 1-3 should be superscript.

Reply:Thank you for your detailed suggestions. We have made changes and improvements according to the submission requirements.

(5) Discussion:

Line 224-225: Delete the sentence “maximum yield at 70,000 plants ha-1 “

Reply:Yes, you are right and we have delete this sentence.

(6) Materials and methods

Please cite two references in 4.2 and 4.4.

Reply:According to your suggestion, we have added two references to better support Materials and methods.

Please pay attention to the case of the first letter of the English word in the sentence. Such as ”4.7. Determination of radiation and accumulated temperature Utilization”.

Reply:

Line 344-345: Figure 7 should be changed to Figure 9.

Reply:We have changed this mistake and thanks a lot.

(7)References need to be uniformly revised and improved according to the “Plants”.

Reply:Thank you for your suggestion. We have revised and improved it according to the requirements of submission.

Round 2

Reviewer 1 Report (New Reviewer)

The author has revised his manuscript carefully. 

This manuscript is a resubmission of an earlier submission. The following is a list of the peer review reports and author responses from that submission.

Round 1

Reviewer 1 Report

1. The basic idea of this research is interesting and the article deals with an important issue.

2. I recommend significantly modifying and supplementing the article, and accordingly also revising the Abstract so that it corresponds to the newly edited article (see below).

3. The article is overall chaotic and partially unclear. Was it possible to eliminate some factors and how was the significance of tested criteria and its influence determined? How these effects are evaluated statistically is unclear in the article.

4. The content of the Introduction section is insufficient, only Chinese literature is cited. I recommend preparing an overview of cultivation practices and yields in the world.

5. Justify why this area of China was chosen. It would be useful to include a map and mark the area it affects and compare it with other areas.

6. Analyze in more detail and separately the influence of the climatic conditions and mainly of water precipitation and soil moisture at the time of the experiments and compare it with the usual conditions at the place of the experiment.

In the Results section:

7. Figures 1, 2, 3, 4 are very confusing and insufficiently explained. In the figures, mark the sub-parts of the graphs as (a), (b), (c), etc.

8. Figure 6 is interesting, but it is not clear if it is an image created by the authors and why it is included in the results.

9. Prepare the Materials and methods section in more details, the text section is confusing.

10. Calculation equations 1, 2, 3, etc. should all be written according to the recommendations for the authors and check the correctness of their notation.

11. Correct many minor formal deficiencies and typos, e.g. on line 310, 320 and 336 there should be a space between the number and the unit, e.g. instead of 30cm it should be 30 cm, etc.

12. Figure 7 is unclear. I recommend reworking.

13. In the Discussion section, compare your results with the results of the works of other authors, cited in the newly supplemented and revised Introduction section.

14. Conclusions completely rework and focus only on the results of your own research.

Author Response

Comments 1 and 2: The basic idea of this research is interesting and the article deals with an important issue. I recommend significantly modifying and supplementing the article, and accordingly also revising the Abstract so that it corresponds to the newly edited article (see below).

Reply: Thanks for being the reviewer this manuscript and helpful suggestions, which comments are very important to revised the paper. We have made a great effort to revise the manuscript in response to these comments and suggestions, and made some revisions in the Abstract with red bold font words.

Comments 3: The article is overall chaotic and partially unclear. Was it possible to eliminate some factors and how was the significance of tested criteria and its influence determined? How these effects are evaluated statistically is unclear in the article.

Reply: First, thank you for your attention and question, which is also an important basis for the topic selection of this paper. It is true that fertilizer, density, and row spacing practices have been reported in the past as the key cultivation techniques for increasing spring maize yield. Meanwhile, in the process of research, we found that previous reports only emphasized the single measure or measure optimization, without considering the priority sequence among this measures,When the production conditions are fully met, we can combine multiple measures to achieve the yield target, but when the conditions are not sufficient, we can only choose one or two cultivation measures, how to choose to reduce the loss of yield, there is still a lack of relevant research reports,and which is a common problem in spring maize region in cold region with relatively deficient production conditions. Therefore, based on the optimal cultivation model, we set up treatments to eliminate key cultivation measures respectively, and measured the contribution of fertilizer, density and row spacing to yield by the loss of yield. This is not based on statistical models, but we have obtained credible and consistent data through three years of field yield data to support our findings. Therefore, we hope to publish it to provide theoretical basis and technical reference for improving the efficient production of spring maize in cold regions.

Comments 4: The content of the Introduction section is insufficient, only Chinese literature is cited. I recommend preparing an overview of cultivation practices and yields in the world.

Reply:Thank you for your suggestion. In the revised version, we have reedited the Introduction section and added relevant literature about cultivation practices and yield in the world.

Supplementary References list:

  1. Louarn G, Chenu K, Fournier C, Andrieu B, Giauffret C. Relative contributions of light interception and radiation use effi-ciency to the reduction of maize productivity under cold temperatures. Functional Plant Biology 2008, 35(9-10): 885-899.
  2. Liu T, Song F, Liu S, Zhu X. Canopy structure, light interception, and photosynthetic characteristics under different nar-row-wide planting patterns in maize at silking stage. Spanish Journal of Agricultural Research 2011, 9(4): 1249-1261.
  3. Liu T, Song F, Liu S, Zhu X. Light interception and radiation use efficiency response to narrow-wide row planting patterns in maize. Australian Journal of Crop Science 2012, 6(3): 506-513.
  4. Liu X, Wang W-x, Lin X, Gu S-b, Wang D. The effects of intraspecific competition and light transmission within the canopy on wheat yield in a wide-precision planting pattern. Journal Of Integrative Agriculture 2020, 19(6): 1577-1585.
  5. Liu ZJ, Yang XG, Lin XM, Hubbard KG, Lv S, Wang J. Maize yield gaps caused by non-controllable, agronomic, and soci-oeconomic factors in a changing climate of Northeast China. Science Of the Total Environment 2016, 541: 756-764.

Comments 5: Justify why this area of China was chosen. It would be useful to include a map and mark the area it affects and compare it with other areas.

Reply: Thank you for your suggestion. Heilongjiang Province is in the northern region of Jilin maize belt, one of the three golden maize belts in the world, and is the most important spring maize production and planting base in China. The annual maize planting area and total output account for more than 30% of the whole country. Therefore, it is of great significance to continuously enhance the production level of maize in the region. At the same time, the experimental site is in the dominant maize producing areas of Heilongjiang Province, with precipitation of 484mm, effective accumulated temperature of 1515℃ and radiation amount of 2381.6MJ/m2 in the growing season. It has the climatic conditions conducive to yield improvement in Heilongjiang Province, so the experimental site is chosen here.

Comments 6: Analyze in more detail and separately the influence of the climatic conditions and mainly of water precipitation and soil moisture at the time of the experiments and compare it with the usual conditions at the place of the experiment.

Reply: Thank you for your suggestion. In the last part of the “Introduction”, we have included an analysis of climate conditions in the region and compared the experimental years with the average climate conditions of past years.

Comments 7: In the Results section: Figures 1, 2, 3, 4 are very confusing and insufficiently explained. In the figures, mark the sub-parts of the graphs as (a), (b), (c), etc.

Reply: Thank you for your suggestion. In the revised version, we revised the annotations of pictures 1, 2, 3 and 4 and marked each small picture with (a), (b) and (c).

Comments 8: Figure 6 is interesting, but it is not clear if it is an image created by the authors and why it is included in the results.

Reply: Thank you for your question. Yes, Figure 6 is a schematic diagram compiled by the author according to the results of this study. Since it is a summary of all the findings, the author puts it at the end of the results section, which is cited in the summary description paragraph.

Comments 9: Prepare the Materials and methods section in more details, the text section is confusing.

Reply: Thank you for your suggestion. In the revised version, we revised the description of “Materials and Methods” and corrected the complicated sentences, hoping to increase readability.

Comments 10: Prepare the Materials and methods section in more details, the text section is confusing. Calculation equations 1, 2, 3, etc. should all be written according to the recommendations for the authors and check the correctness of their notation.

Reply: Thank you for your suggestion. In the revised version, we revised the description of materials and methods, and revised the writing and marking of the formula.

Comments 11: Correct many minor formal deficiencies and typos, e.g. on line 310, 320 and 336 there should be a space between the number and the unit, e.g. instead of 30cm it should be 30 cm, etc.

Reply: Thank you for your careful review. We have corrected many spelling and typing errors in the revised version and marked the revised part in red for your review.

Comments 12: Figure 7 is unclear. I recommend reworking.

Reply: Thanks for your suggestion. We have modified Figure 7 and replaced it.

Comments 13: In the Discussion section, compare your results with the results of the works of other authors, cited in the newly supplemented and revised Introduction section.

Reply: Thanks for your suggestions. We have edited the “Discussion” section and added some new content and literature, which are listed below for your review.

Comments 14: Conclusions completely rework and focus only on the results of your own research.

Reply: Thank you for your suggestion. We have edited the “Conclusion” again. See the red part for specific modifications

Reviewer 2 Report

Authors presented an interesting concept in the submitted manuscript.

It is hard to read, English needs to be improved. I tried to suggest corrections many places, but there were sentences, that I couldn’t even guess what Authors tried to convey.

The introduction section needs to include little more work from past research. There are lots of available publication related to row spacing (row configuration) plant density, etc. Authors have barely mentioned previous research related to narrow row spacing, or the twin-row row spacing (that would be similar to the narrow-wide rows system used here).

Authors also need to include many more details in the M&M section, to be able to follow what they have done.

Authors have based the main research only on a one year, one location this main research idea (or at least optimizing some of their parameters) – Either this data is not sufficient to make conclusions for these parameters, or not relevant enough to include in the manuscript.

Many of the presented data (in the text) is not supported by their own figures and tables.

Ln 37: What do you mean: ‘it remains has a high potential to achieve fully’? Please clarify the sentence

Ln 46: delete the ‘population’ word

Ln 49-50: What do you mean: The more uniform distribution of radiant could achieve the elevating of underlying density’ Please clarify it

Ln 52: What do you mean: ‘the movement of dry matter after anthesis including’

Ln 53: use the word ‘influenced’ instead of ‘regulated’

Ln 56: what do you mean under ‘unharmonious planting spacing’?

Ln 60: delete ‘population’ – and many other places

Ln 62-64: Clarify this sentence

Ln 69: delete population

Ln 79-80: Figure 1 doesn’t indicate any differences between 70 and 80K density. Seems like the wide-narrow row spacing was beneficial only at the 60 K density. Did you have interaction between the row arrangement and the plant density? Presenting and supporting your results with proper statistics is required.

Ln 84: How can ‘future research’ suggest things? I am not a futurist, or at least can’t see to the future

Ln 86: Based on your Figure 1, I do not see the benefit to increase plant  density beyond 70K? How about economics for the additional seed cost?

Ln 93: what do you mean: ‘then decreases with the plant height’ please clarify it

Figure 2: Use letters for separating the different panels – and you can help the reader to point out in the result presentation

Ln 108: Your graphs showing LAI (according to the axis at least). Correct the caption.

Ln 109: How do you compare the treatments? What are the treatments? Please clarify how to interpret your statistical notation

Ln 115: what is ‘relatively luxuriant’

Ln 118: OM-D did not reduce it

Ln 121 The light transmittance was actually higher in OM-S than in OM (Fig 3) in 2018, and did not see differences in 2019

Figure 3: Use letters for separating the different panels – and you can help the reader to point out in the result presentation

Ln 129: How can this be in 2017?

Figure 4: I thought your omission experiment was conducted in 2018 and 2019. Are the years correct in the graph?

Ln 136: What is ‘transshipment, transshipment rate’ ?

Ln 142: Where is this 8.5% coming from? Where did you presented?

Ln 141 – 145: Table 1 doesn’t support this. Looking your Table 1 there were no statistical differences

Ln 174: Where did you present the 23.85 and 16.05% contribution? I couldn’t calculated from Table 2

Ln 177: Looking Table 3: there were no differences in 100-kernel weight

Ln 178: You did not show any data regarding number of rows and the number of grains per row

Table 3: Your manuscript is working with corn. It should be ear and not spike

Ln 190: Isn’t it 2018? I thought your omission experiment was conducted in 2018 and 2019

Ln 203-204: What is ‘early fulling’ stage?

Ln 227: What is light leakage? Please clarify the sentence

Ln 229: What do you mean under ‘light and heat resources’?

Ln 235: is it kernel or ear? Please use the terms correctly – this is corn

Ln 235-236: Please clarify: ‘and the nutrient release of field soil’

Ln 236: I would use like this: ‘Increasing the planting density from 60,000 to 80,000 plants ha-1 in Northeast China, …’

Ln 251-252: You did not have differences between 100-grain weight, and you did not present data about row numbers

Ln 280-282: Please include coordinates for your field locations

Ln 287: What is ‘split-zone design’ look like? I have never heard about this experimental design

Ln 292:

What do you mean ‘seeds were planted with handle’?

Ln 301: How is Figure 8 relevant to this sentence?

Ln 327-329: LAI shouldn’t be the sum of LA of the 5 (or more) leaves? Please clarify and extend on your measurement and calculation method

Ln 352: ‘nitrogen translocation amount(g), nitrogen translocation efficiency (%), nitrogen translocation contribution to grain nitrogen accumulation amount (%)’ N translocation from where?   You have used different zone and whole plants in the results. Please be clear where, how, and what did you measure and calculated. From this description I would not be able to reproduce

Ln 357: delete ‘population’ word

Ln 361: We typically express grain yield at the 15.5% moisture level

Ln 368: How did you calculate the GDDs?

Author Response

Comments 1: Authors presented an interesting concept in the submitted manuscript.

It is hard to read, English needs to be improved. I tried to suggest corrections many places, but there were sentences, that I couldn’t even guess what Authors tried to convey.

Reply: Thanks for being the reviewer this manuscript and helpful suggestions, which comments are very important to revise the paper. We had invited the experts in the agriculture to reviewer the language of the modified version to make it easier to read.

Comments 2: The introduction section needs to include little more work from past research. There are lots of available publication related to row spacing (row configuration) plant density, etc. Authors have barely mentioned previous research related to narrow row spacing, or the twin-row row spacing (that would be similar to the narrow-wide rows system used here).

Reply: Thanks for your suggestions, we have edited the Introduction and added some new contents and literature, which are listed below for your review.

Supplementary References list:

  1. Louarn G, Chenu K, Fournier C, Andrieu B, Giauffret C. Relative contributions of light interception and radiation use effi-ciency to the reduction of maize productivity under cold temperatures. Functional Plant Biology 2008, 35(9-10): 885-899.
  2. Liu T, Song F, Liu S, Zhu X. Canopy structure, light interception, and photosynthetic characteristics under different nar-row-wide planting patterns in maize at silking stage. Spanish Journal of Agricultural Research 2011, 9(4): 1249-1261.
  3. Liu T, Song F, Liu S, Zhu X. Light interception and radiation use efficiency response to narrow-wide row planting patterns in maize. Australian Journal of Crop Science 2012, 6(3): 506-513.
  4. Liu X, Wang W-x, Lin X, Gu S-b, Wang D. The effects of intraspecific competition and light transmission within the canopy on wheat yield in a wide-precision planting pattern. Journal Of Integrative Agriculture 2020, 19(6): 1577-1585.
  5. Liu ZJ, Yang XG, Lin XM, Hubbard KG, Lv S, Wang J. Maize yield gaps caused by non-controllable, agronomic, and soci-oeconomic factors in a changing climate of Northeast China. Science Of the Total Environment 2016, 541: 756-764.

Comments 3: Authors also need to include many more details in the M&M section, to be able to follow what they have done.

Reply: Thanks for your suggestions. We have revised the Materials and methods section and added some new contents and documents, which are listed below for your review.

Comments 4: Authors have based the main research only on a one year, one location this main research idea (or at least optimizing some of their parameters) – Either this data is not sufficient to make conclusions for these parameters, or not relevant enough to include in the manuscript.

Reply: Thank you for your question. In the subsequent de-factor study, the close-planting and high-yield cultivation mode selected in this study was repeated as a control treatment for two years, and the results of the three years of research data were consistent. This kind of multi-measures did achieve the close-planting and high-yield of spring maize. Therefore, combined with the reality of relatively deficient cultivation and production management of spring maize in this region, we further set up experimental treatments by eliminating cultivation measures to explore the contribution of each cultivation measure to yield and its mechanism.

Comments 5: Many of the presented data (in the text) is not supported by their own figures and tables.

Reply: Thank you for your question. We have checked the charts in the article first and revised some paragraphs of the results to make them easier to understand.

Comments 6: Ln 37: What do you mean: ‘it remains has a high potential to achieve fully’? Please clarify the sentence.

Reply: What the author wants to convey is that the current grain yield is only 70 percent of the potential yield, and there is still a potential yield to be tapped.

Comments 7: Ln 46: delete the ‘population’ word.

Reply: Thanks for your suggestion, we have removed the word "population" in the modified version.

Comments 8: Ln 49-50: What do you mean: The more uniform distribution of radiant could achieve the elevating of underlying density’ Please clarify it.

Reply: Thank you for your question. The message we want to convey is " The relatively uniform radiation distribution in the canopy can greatly reduce the competition of individual plants in the population for radiation, and improve the population capacity and potential density."

Comments 9: Ln 52: What do you mean: ‘the movement of dry matter after anthesis including’.

Reply: Thank you for your question. The message we want to convey is " The transport and distribution of dry matter from vegetative organs to reproductive organs after anthesis includes the production and distribution of photosynthate".

Comments 10: Ln 53: use the word ‘influenced’ instead of ‘regulated’.

Reply:Thanks to your suggestion, we replaced "regulated" with "influenced" in the modified version.

Comments 11: Ln 56: what do you mean under ‘unharmonious planting spacing’?

Reply: Thank you for your question, this sentence we want to convey the awareness of "ununiform planting row spacing".

Comments 12: Ln 60: delete ‘population’ – and many other places.

Reply: Thanks for your suggestion. We have removed the "population" of Ln60 and some other inappropriate places in the modified version.

Comments 13: Ln 62-64: Clarify this sentence.

Reply: Thank you for your question. What we want to convey in this sentence is that China is the country with the largest nitrogen fertilizer consumption in the world, and the nitrogen fertilizer utilization efficiency is 50% lower than that of the major countries in the world. This is due to the abuse of nitrogen fertilizer in the major corn production areas in China.

Comments 14: Ln 69: delete population.

Reply: Thanks for your suggestion. We have deleted the "population" of Ln69 and some other inappropriate places in the revised version.

Comments 15: Ln 79-80: Figure 1 doesn’t indicate any differences between 70 and 80K density. Seems like the wide-narrow row spacing was beneficial only at the 60 K density. Did you have interaction between the row arrangement and the plant density? Presenting and supporting your results with proper statistics is required.

Reply: In Fig. 1, the yield data of 70K and 80K did not have statistically difference, but compared with 70K, 80K significantly increased the planting density by 14.3%. According to the regression analysis of yield on planting density, it also showed that the inflection point of the regression line intersected at 80K planting density, and the planting with equal row spacing had a similar change trend.  The highest yield occurred at the planting density of 70K. In order to achieve high yield in dense planting, we selected 80K, which was a higher planting density, as the planting density of dense planting and increased yield cultivation mode.

Comments 16: Ln 84: How can ‘future research’ suggest things? I am not a futurist, or at least can’t see to the future.

Reply: Thank you for your question. The message we want to convey is "further research findings."

Comments 17: Ln 86: Based on your Figure 1, I do not see the benefit to increase plant density beyond 70K? How about economics for the additional seed cost?

Reply: Thank you for your question. Increasing planting density is an important way to improve spring corn yield, which has been agreed by many experts in the world. Meanwhile, in this study, the maximum yield of 80K reached 12179.24kg/hm2 in different replicates. The statistical results showed that there was no difference in the difference of soil fertility on average, but the increase of yield was considerable when the local fertility conditions were fully met. Therefore, 80K was selected as the planting density of the optimal mode based on this.

Comments 18: Ln 93: what do you mean: ‘then decreases with the plant height’ please clarify it.

Reply: Thank you for your question. What we want to convey in this sentence is the awareness that "there will be a decreasing trend with the increase of plant height".

Comments 19: Figure 2: Use letters for separating the different panels – and you can help the reader to point out in the result presentation.

Reply: Thank you for your suggestion. In the revised version, we will label each small picture with (a), (b), (c)

Comments 20: Ln 108: Your graphs showing LAI (according to the axis at least). Correct the caption.

Reply: Thank you for your suggestion. We have revised this part of the description in the revised version.

Comments 21: Ln 109: How do you compare the treatments? What are the treatments? Please clarify how to interpret your statistical notation.

Reply: Thank you for your questions, we are the same leaf layer of different leaf area index were compared between the test processing, at the time of drawing react differently to image processing, the vertical distribution of leaf area index will be treated as the same in different leaf layers in a picture, but the letters marked differences significant characterization is the difference between different processing with leaf layer.

Comments 22: Ln 115: what is ‘relatively luxuriant’.

Reply: Thank you for your question. What we want to convey in this sentence is "the dry weight of leaves is relatively large".

Comments 23: Ln 118: OM-D did not reduce it.

Reply: Thank you for your question. In the process of revising the manuscript, we re-calculated and counted this part of data, and found some inappropriate sentences. Once they were modified in the revised version, they were marked in red for your review.

Comments 24: Ln 121 The light transmittance was actually higher in OM-S than in OM (Fig 3) in 2018, and did not see differences in 2019.

Reply: Thank you for your review, we calculate this part of the data, found that there are have differences in 2019 but only at 150cm leaf layer was reached statistically level, but the 2018 data good shows the difference between the two treatments, based on this the description of the results has been reedited and reded in font for your review.

Comments 25: Figure 3: Use letters for separating the different panels – and you can help the reader to point out in the result presentation

Reply: Thank you for your suggestion. In the revised version, we will label each small picture with (a), (b), (c)

Comments 26: Ln 129: How can this be in 2017?

Reply: Thank you for your care. We completed this study from 2017 to 2019, which is a complete repetition of the experiment in three years. However, due to the annual workload, we chose to complete part of the indicators in 2017-2018, and some indicators in 2018-2019. The data presented in Fig. 4 are the samples obtained in 2017 and 2018, and the data measurements completed in 2018 and 2019, so we mark the time as the year of sample acquisition.

Comments 27: Figure 4: I thought your omission experiment was conducted in 2018 and 2019. Are the years correct in the graph?

Reply: Thank you for your care. We completed this study from 2017 to 2019, which is a complete repetition of the experiment in three years. However, due to the annual workload, we chose to complete part of the indicators in 2017-2018, and some indicators in 2018-2019. The data presented in Fig. 4 are the samples obtained in 2017 and 2018, and the data measurements completed in 2018 and 2019, so we mark the time as the year of sample acquisition.

Comments 28: Ln 136: What is ‘transshipment, transshipment rate’ ?.

Reply: In this case, the amount of nitrogen accumulation was calculated from the product of nitrogen content of individual organs and dry matter weight, and then the amount of grain filling and the transfer rate were calculated from the difference between the silking stage and the maturity stage.

Comments 29: Ln 142: Where is this 8.5% coming from? Where did you presented?

Reply: Here 8.5% was calculated from the difference between the "OM" and "OM-D" stem translocation rates, which were recalculated in the modified version and corrected for input errors.

Comments 30: Ln 141 – 145: Table 1 doesn’t support this. Looking your Table 1 there were no statistical differences.

Reply: Thank you for your advice, indeed, in statistics, there were no significant differences between the data but real existence in number is different, may be due to the large difference between repeat cover the differences between the treatment, but here we just want to present different organs between nitrogen accumulation and migration of response to the different cultural practices, because we believe that,  Nitrogen accumulation and transport are important components of yield differential formation.

Comments 31: Ln 174: Where did you present the 23.85 and 16.05% contribution? I couldn’t calculated from Table 2.

Reply: Thank you for asking, because this data is presented in table3.

Comments 32: Ln 177: Looking Table 3: there were no differences in 100-kernel weight.

Reply: Yes, there was no significant difference in 100-grain weight in this study, which may be because we used the averages of each years as repeaters in the statistical analysis of this part of data, because the inter-annual influence on this trait covered up the error between treatments, but there was still a big difference between the averages.

Comments 33: Ln 178: You did not show any data regarding number of rows and the number of grains per row.

Reply: Yes, due to the space problem, we did not present the data of the number of rows and grains in rows, but the number of grains per ear was calculated according to the number of rows per ear and grains per row. In order to better explain our results, we added this part of data in the revised version.

Comments 34: Table 3: Your manuscript is working with corn. It should be ear and not spike.

Reply: Thank you for your suggestion. We have corrected the response part in the modified version.

Comments 35: Ln 190: Isn’t it 2018? I thought your omission experiment was conducted in 2018 and 2019.

Reply: Thank you for your care. We completed this study from 2017 to 2019, which is actually a complete repetition of the experiment in three years. However, due to the annual workload, we chose to complete part of the indicators in 2017-2018, and some indicators in 2018-2019.  The data presented in FIG. 4 are the samples obtained in 2017 and 2018, and the data measurements completed in 2018 and 2019, so we mark the time as the year of sample acquisition.

Comments 36: Ln 203-204: What is ‘early fulling’ stage?.

Reply: Thank you for your question, this sentence we want to convey the consciousness is "fulling stage" We have made corresponding changes in the revised version.

Comments 37: Ln 227: What is light leakage? Please clarify the sentence.

Reply: Thank you for your question. The message we want to convey is "radiation losing." We have made corresponding changes in the revised version.

Comments 38: Ln 229: What do you mean under ‘light and heat resources’?

Reply: Thank you for your question. The message we want to convey is " radiation and accumulated temperature."

Comments 39: Ln 235: is it kernel or ear? Please use the terms correctly – this is corn.

Reply: Thank you for your question. The message we want to convey is "light and heat resources." We have made corresponding changes in the revised version.

Comments 40: Ln 235-236: Please clarify: ‘and the nutrient release of field soil’.

Reply: Thank you for your question. The message we want to convey is "soil nutrient ". We have made corresponding changes in the revised version.

Comments 41: Ln 236: I would use like this: ‘Increasing the planting density from 60,000 to 80,000 plants ha-1 in Northeast China, …’.

Reply: Thank you for your suggestion. We have revised the relevant description in the revised version

Comments 42: Ln 251-252: You did not have differences between 100-grain weight, and you did not present data about row numbers.

Reply: Thank you for your suggestion. We have revised the relevant description in the revised version

Comments 43: Ln 280-282: Please include coordinates for your field locations.

Reply: Thank you for your suggestion. We have revised the relevant description in the revised version

Comments 44: Ln 287: What is ‘split-zone design’ look like? I have never heard about this experimental design.

Reply: Thank you for your question, but the message we're trying to convey is "split-zone design."

Comments 45: Ln 292: What do you mean ‘seeds were planted with handle’? Reply: Thank you for your question. The message we want to convey is "seeds were sowed by handle".

Comments 46: Ln 301: How is Figure 8 relevant to this sentence?

Reply: Thanks for your suggestion. In the revised version, we have added the relevant description to the last paragraph of the preface.

Comments 47: Ln 327-329: LAI shouldn’t be the sum of LA of the 5 (or more) leaves? Please clarify and extend on your measurement and calculation method.

Reply: Thanks for your suggestion. We have edited the Materials and Methods section and revised the description of this part

Comments 48: Ln 352: ‘nitrogen translocation amount(g), nitrogen translocation efficiency (%), nitrogen translocation contribution to grain nitrogen accumulation amount (%)’ N translocation from where?   You have used different zone and whole plants in the results. Please be clear where, how, and what did you measure and calculated. From this description I would not be able to reproduce.

Reply: Thanks for your suggestion. We have edited the Materials and Methods section and revised the description of this part

Comments 49: Ln 357: delete ‘population’ word.

Reply: Thanks for your suggestion. We have removed the "population" of Ln357 and some other inappropriate places in the revised version.

Comments 50: Ln 361: We typically express grain yield at the 15.5% moisture level.

Reply: Thanks for your suggestion. We have recalculated the grain yield and adjusted the water content to 15.5%

Comments 51: Ln 368: How did you calculate the GDDs?

Reply: We first obtained the daily average temperature, and summed the average temperature higher than 10℃ in the growing season to obtain the accumulated temperature. In the modified version, we added the response calculation formula and references.

Round 2

Reviewer 1 Report

Minor formal deficiencies need to be corrected everywhere, e.g. on line 72 there should be a space between the number and the unit, e.g. instead of 484mm it should be 484 mm. Similarly, in other parts of the article.

All typos need to be corrected, e.g. on line 75, it is misspelled Famers, correct to Farmers. Similarly, correct all typos and inaccuracies of the English language in all parts of the article.

Author Response

Dear Reviewer 1,

As you said, due to our mistakes, the above-mentioned problems do exist in the manuscript. We are very sorry that our laxity in language and grammar has caused difficulties in reading your manuscript. According to your suggestion, we have thoroughly revised and improved the spelling, grammar and layout of the manuscript. Thank you very much for your valuable suggestions. Our scientific research team will follow your suggestions. In future manuscript writing, teachers and students will participate together, especially pay attention to details, so as to develop good scientific research writing habits and eye-catching working attitude. Thank you very much for your very useful suggestions on the improvement of the manuscript. If you have any other questions, please feel free to contact me.

Best reagrds,

Lin Piao, Shiyu Zhang, Junyao Yan, Tianxu Xiang, Yang Chen, Ming Li*, and Wanrong Gu*

6 September, 2022

Reviewer 2 Report

After reviewing the resubmitted manuscript, I still recommend to release the manuscript.

It was still really difficult to read, follow, and understand the manuscript. You must re-write the entire manuscript using proper English grammar.

Your statistical description and result presentation is inconsistent. You stated that your studies were conducted between 2017 and 2019. Why did you just selectively show result from 2017 and 2018, and some cases from 2018 and 2019, and even at some point from 2019 only (NUE calculation)?  Were you presenting your data across the three years when you did not indicate years?

What was the reason for not showing research data from all three years? Why did you combine years for certain data and not for others? All this should be included in your M&M and statistical description. How did you handle the year in your statistical model(s)?

Did you do regression analysis to determine the optimal seeding rate? Did you statistically compare the uniform row spacing vs. narrow-wide row spacing in your population study? You have to show all those details before you just make a statement.

Author Response

Dear Reviewer 2,

Thank you very much for your series of questions. Our team very much agrees with your modification suggestions. We will reply and explain from the following aspects.

Firstly, the research aimed to assess the contribution of fertilizer, density, and row spacing in the integrated cultivation measures and identify their regulation mechanism on canopy architecture and factors in biomass accumulation in spring maize. Fertilizer, density, and row spacing practices have been reported in the past as the key cultivation techniques for increasing spring maize yield. Meanwhile, in the process of research, we found that previous reports only emphasized the single measure or measure optimization, without considering the priority sequence among this measures,When the production conditions are fully met, we can combine multiple measures to achieve the yield target, but when the conditions are not sufficient, we can only choose one or two cultivation measures, how to choose to reduce the loss of yield, there is still a lack of relevant research reports,and which is a common problem in spring maize region in cold region with relatively deficient production conditions. Therefore, based on the optimal cultivation model, we set up treatments to eliminate key cultivation measures respectively, and measured the contribution of fertilizer, density and row spacing to yield by the loss of yield. This is not based on statistical models, but we have obtained credible and consistent data through three years of field yield data to support our findings. Therefore, we hope to publish it to provide theoretical basis and technical reference for improving the efficient production of spring maize in cold regions. In the subsequent de-factor study, the close-planting and high-yield cultivation mode selected in this study was repeated as a control treatment for two years, and the results of the three years of research data were consistent. This kind of multi-measures did achieve the close-planting and high-yield of spring maize. Therefore, combined with the reality of relatively deficient cultivation and production management of spring maize in this region, we further set up experimental treatments by eliminating cultivation measures to explore the contribution of each cultivation measure to yield and its mechanism.

Secondly, the problem that our scientific research team needs to explain is that the experiments in recent years are not on the same test site, because they are affected by force majeure factors. Due to the personnel change and development adjustment of Northeast Agricultural University, the original experimental plot was expropriated, so our scientific research team had to change the experimental plot, which is also a situation we don't want to see. We have been striving to preserve the experimental plot, but we are an ordinary maize scientific research worker, so there is nothing we can do. At the same time, we need to note that although the test site has been changed, the results and conclusions of the test are absolutely correct and can stand scrutiny. In fact, when we were writing this manuscript, some experts suggested that it would be better to write it as a unified experimental plot. However, based on our scientific, serious and realistic attitude, we will truthfully write down the experiments we do, without any false or untenable results.

After writing the manuscript, we also consulted relevant experts and asked them whether such experimental design and experimental processing between years were feasible? These experts said that they have always recognized that as long as the results and designs are based on objective facts, they can be recognized. They also said that many well-known journals in the industry such as 《Field crops research》,《European journal of agronomy》 have published many results similar to experimental design.

Thirdly, after receiving the email notification from the editorial department last night, the experts of our scientific research team carefully studied your suggestions and questions through Tencent video conference software. We think we need to reiterate the purpose of our experimental design, so it is also important to include the wide and narrow rows of maize and the density of maize mentioned by you, but it is not the focus of our research content. We mainly discuss it from the perspective of "contribution", not the effect of individual technical measures. We look forward to your understanding and recognition.

Fourthly, our scientific research team has carried out the above research for many years. We have also actively cooperated with Chinese Academy of Agricultural Sciences, Jilin Academy of Agricultural Sciences, Shandong Agricultural University, Northwest University of agriculture and forestry science and technology and other universities and scientific research colleagues, and made a series of progress. The postdoctoral of our research group, namely the first author of the manuscript, Piao Lin, also focuses on the contribution rate of comprehensive agricultural measures in her postdoctoral research work, and has published many articles in well-known journals in the industry. During her post doctoral period, she also received a grant from the China Post Doctoral Fund. The research content of this draft is one of the results of a series of studies. At present, she is facing the critical moment of post doctoral exit, and we look forward to your support.

At a time when scientific research is becoming more and more convoluted and when the epidemic is raging around the world, our discipline team also believes that there are inevitably small problems in the experimental design of this draft. Despite this, we have been doing something in this field. In fact, we have also made a series of innovative achievements. In the process of sorting out and publishing one after another, many articles have been submitted to 《Field crops research》. We look forward to your further support and encouragement.

Fifthly, we have not submitted more than one manuscript. All the authors are experts and graduate students in the field of maize cultivation. All the authors agree with the contents of the manuscript. The article conforms to the code of ethics and is also allowed by relevant organizations. We are in a worried mood. We do not know whether the above reply can satisfy you, but our team has been looking forward to receiving your further guidance and reply, and especially looking forward to giving us another opportunity to modify and improve. In addition, we need to explain again that due to the priority of space and the gradual nature of scientific research, no work can be done comprehensively and completely in one manuscript, so our scientific research team will supplement and improve it in the next scientific research and manuscript writing. Thank you very much for your very useful suggestions on the improvement of the manuscript. If you have any other questions, please feel free to contact me.

Best reagrds,

Lin Piao, Shiyu Zhang, Junyao Yan, Tianxu Xiang, Yang Chen, Ming Li*, and Wanrong Gu*

6 September, 2022